# Use of Multivariate Adaptive Regression Splines Algorithm to Predict Body Weight from Body Measurements of Anatolian buffaloes in Türkiye

**DOI:** 10.3390/ani12212923

**Published:** 2022-10-25

**Authors:** Oğuz Ağyar, Cem Tırınk, Hasan Önder, Uğur Şen, Dariusz Piwczyński, Esra Yavuz

**Affiliations:** 1Department of Veterinary, Kahta Vocational School, Adıyaman University, Adıyaman 02400, Turkey; 2Department of Animal Science, Faculty of Agriculture, Igdir University, Iğdır 76000, Turkey; 3Department of Animal Science, Ondokuz Mayis University, Samsun 55139, Turkey; 4Department of Agricultural Biotechnology, Ondokuz Mayis University, Samsun 55139, Turkey; 5Department of Animal Biotechnology and Genetic, Faculty of Animal Breeding and Biology, Bydgoszcz University of Science and Technology, 85-796 Bydgoszcz, Poland:; 6Department of Accounting and Tax Practices, Çizre Vocational School, Şırnak University, Şırnak 73200, Turkey

**Keywords:** water buffalo, MARS, prediction, biometric properties

## Abstract

**Simple Summary:**

The present study was performed to estimate body weight (BW) from several body measurements, such as tail length (TL), shoulder height (SH), withers height (WH), body length (BL), chest circumference (CC), shank diameter (SD) and birth weight (BiW). The data set was taken from Muş Province of Türkiye. In this respect, 171 Anatolian buffaloes were used. To estimate the BW, different proportions of the training and test sets were used with the MARS (Multivariate Adaptive Regression Splines) algorithm. In conclusion, it could be suggested that the MARS algorithm may allow animal breeders to obtain an elite population and to determine the body measurements affecting BW as indirect selection criteria for describing the breed description of Anatolian buffalo and aiding sustainable meat production and rural development in Türkiye.

**Abstract:**

Anatolian buffalo is an important breed reared for meat and milk in various regions of Türkiye. The present study was performed to estimate body weight (BW) from several body measurements, such as tail length (TL), shoulder height (SH), withers height (WH), body length (BL), chest circumference (CC), shank diameter (SD) and birth weight (BiW). The data set was taken from Muş Province of Türkiye. In this respect, 171 Anatolian buffaloes were used. To estimate the BW, different proportions of the training and test sets were used with the MARS algorithm. The optimal MARS was determined at a proportion of 70–30%. The MARS model displays the heaviest BW that can be produced by Anatolian buffalo according to tail length, body length, chest circumference and shoulder height. In conclusion, it could be suggested that the MARS algorithm may allow animal breeders to obtain an elite population and to determine the body measurements affecting BW as indirect selection criteria for describing the breed description of Anatolian buffalo and aiding sustainable meat production and rural development in Türkiye.

## 1. Introduction

Domestic buffalo, called “Water Buffalo”, belonging to the species Bubalus bubalis, is divided into two different types, namely river and swamp buffalo. While swamp buffaloes generally benefit from draft power and meat yield, river buffaloes are valuable in terms of both meat and milk yield characteristics [1]. Buffaloes, which are considered to be of economic importance in terms of milk and meat yield among cattle, are grown in Asia, South America, North Africa, all Mediterranean countries except France, some Central European countries and Australia [2,3]. Buffalo breeding has advantages, such as their high resistance to natural conditions and diseases, high feed conversion rates and lower cost compared to breeding cattle [4,5]. Anatolian buffaloes are classified as a river type belonging to the Mediterranean group within the water buffalo group [1,6,7]. According to the literature, the existence of the buffalo species in Türkiye is thought to date back to 3000 BC [8]. In 2021, there were 186,000 head buffaloes in Türkiye according to the TUIK database [9].

Anatolian buffaloes are covered with black-gray skin and sparse feathers. The horns of the Anatolian buffalo are in the form of an arc. Calves are naturally black at birth. However, they turn reddish-brown at 3–6 months of age. At the age of 10–12 months, they turn black again. Anatolian buffaloes are very resistant to unexpected changes in feed and traditional diseases [8].

Body weight is the most important economic characteristic of all meat animals because farmers’ incomes are directly derived from the animal’s weight. In recent years, more attention has been paid to explaining the relationship between body size and body weight to increase meat production. Live weight estimations based on body measurements in buffaloes can reveal growth and development characteristics as well as production performance and genetic characteristics of buffaloes. In recent years, several body measurements taken during early growth periods have been used as early selection criteria to improve the relative proportion of superior buffalo offspring with good body weight in future populations [10]. It has also been reported that these measurements are practically helpful for buffalo breeders willing to estimate body weight, which is essential for herd management.

In rural conditions where there is no scale, buffalo body measurements and body weight estimation contribute to the determination of the optimum feed amount per buffalo in the herd, the marketing price, the dose of medicinal drug to be used in the treatment of diseases and the optimum slaughter time [10]. In this context, estimation based on body measurements, which is used as one of the positive effects on live weight, is accepted as an indirect selection criterion in buffalo breeding. In this framework, reliable techniques such as data mining algorithms and multivariate statistical methods are used to perform phenotypic species identification of buffaloes as the best way to determine effective body measurements.

Many studies have been published on estimating body weight from body measurements in different animal species, such as sheep, cattle, rabbits, dogs and camels [10,11,12,13,14]. However, there is no such study for the Anatolian buffalo breed.

The information on body weight prediction in Anatolian buffaloes by body measurements is insufficient. The prediction is of great importance for making much better decisions on flock management, breed standards, breeding schemes and conserving gene reserves of Anatolian buffaloes. In this respect, sophistical statistical techniques can help to produce more reliable estimates within the scope of indirect selection criteria to be applied in buffalo and to reveal body measurements that affect body weight. To our knowledge, no documented information is available on employing the MARS algorithm for the body weight prediction of Anatolian buffaloes. The current study has been carried out both to fill this gap in the literature and compare these algorithms’ prediction performances.

## 2. Materials and Methods

In this study, the Anatolian buffalo was used as a subject. A total of 171 buffaloes (99 female and 72 male animals) were provided by a private farm in Muş Province of Türkiye.

To predict body weight, body measurements were taken from the buffalo data set for buffaloes 10 to 17 months of age, including tail length (TL), shoulder height (SH), withers height (WH), body length (BL), chest circumference (CC), shank diameter (SD), birth weight (BiW) and body weight (BW).

The multivariate Adaptive Regression (MARS) algorithm was proposed by Friedman [15] for making predictions based on quantitative features. To solve regression-type problems, the MARS algorithm uses a non-parametric regression procedure that allows better recognition of linear, nonlinear and interaction effects between response and explanatory variables. The MARS algorithm was developed from the CART algorithm. In solving regression-type problems, there is no need for any assumptions about both the distribution of variables and the relationships between response and explanatory variables in this algorithm [16,17]. The algorithm has various slopes in the training data set, splitting up the individual segmented linear segments (splines) [16]. The splines relate without problems and form connection points called “knots”. Candidate nodes are randomly placed within the range of each estimator, so the model estimation made with the MARS algorithm is more flexible and interpretable with the help of piecewise linear regressions.

The algorithm generates basic functions based on a step-by-step procedure, considering all probable interaction effects between candidate knots and explanatory variables. The algorithm has two stages, forward and backward passes [18].

The initial stage of this algorithm is the forward pass stage. In this phase, the algorithm begins with an intercept in the initial model. It iteratively contains the basic function sets with the smallest amount of training error to enhance the model. The forward pass stage naturally produces an overfitted model that reaches maximum complexity. The model created from the forward pass stage has a particularly good fit. However, its generalization capability can be inadequate for a data set before an indeterminate structured model is applied, which means an overfitting crisis. The basic functions in the forward pass stage that present the smallest amount to the prediction model are reduced in the backward pass stage, and this situation is used to solve this problem.

The expression used by the algorithm to estimate body weight from explanatory variables that affect body weight, such as body measurements, is given below:$$\hat{y}=\beta_{0}+\overset{M}{\underset{m=1}{\sum }}\beta_{m}\overset{K_{m}}{\underset{k=1}{\prod }}h_{\mathrm{km}}\left(X_{v\left(k,m\right)}\right)$$
where $\hat{y}$ is the predicted value for BW, β_0_ is an intercept, β_m_ is the basic function coefficient, K_m_ is the parameter that limits the interaction order, the h_km_ (X_v(k,m)_) term is called the basis function, and v(k,m) is an index of the explanatory variable in the m^th^ component of the k^th^ product. The basic functions that reduce the performance of the model obtained after the forward and backward pass stages are eliminated due to the generalized cross-validation error (GCV) [11,19]. The formula for the GCV is given below:$$\mathrm{GCV}\left(\lambda \right)=\frac{\sum_{i=1}^{n}{\left(y_{i}-y_{\mathrm{ip}}\right)}^{2}}{{\left[1-\frac{M\left(\lambda \right)}{n}\right]}^{2}}$$
where: n is the size of the training set, y_i_ is the observed value for the response variable (BW), y_ip_ is the estimated value for the response variable (BW), and M(λ) is the penalty term for the complexity of the model that includes the λ terms.

At the first stage of the analysis, the multicollinearity between the explanatory variables was examined, and there was no multicollinearity problem for the data set according to the variance inflation factor (VIF) results. To predict BW utilizing the training data set, the 10-fold cross-validation procedure was utilized to decide the optimum MARS model.

To evaluate the performance of the MARS algorithm, the following goodness-of-fit criteria were used [11,19,20]:
Root mean square error (RMSE):$$\mathrm{RMSE}=\sqrt{\frac{1}{n}\overset{n}{\underset{i=1}{\sum }}{\left(y_{i}-y_{\mathrm{ip}}\right)}^{2}}$$Akaike information criterion (AIC):$$\left\{\begin{array}{c} AIC=n.ln\left[\frac{1}{n}\overset{n}{\underset{i=1}{\sum }}{\left(y_{i}-y_{ip}\right)}^{2}\right]+2k,\;\;\;\;\;if\;n/k>40\;\\ AIC_{c}=AIC+\frac{2k\left(k+1\right)}{n-k-1}\;\;\;\;\;\;\;\;\;\;\;\;\;\;\;\;\;\;\;\;\;\;\;\;\;\;\;\;\;\;\;\;\;\;\;\;\;otherwise \end{array}\right.$$Standard deviation ratio (SDR):$${\mathrm{SD}}_{\mathrm{ratio}}=\frac{S_{m}}{S_{d}}$$Performance index (PI):$$\mathrm{PI}=\frac{\mathrm{rRMSE}}{1+r}$$Global relative approximation error (RAE):$$\mathrm{RAE}=\sqrt{\frac{\sum_{i=1}^{n}{\left(y_{i}-y_{\mathrm{ip}}\right)}^{2}}{\sum_{i=1}^{n}y_{i}{}^{2}}}$$Mean absolute percentage error (MAPE):$$\mathrm{MAPE}=\frac{1}{n}\;\overset{n}{\underset{i=1}{\sum }}\left|\frac{y_{i}-y_{\mathrm{ip}}}{y_{i}}\right|*100$$Mean absolute deviation (MAD):$$MAD=\frac{1}{n}\;\overset{n}{\underset{i=1}{\sum }}\left|y_{i}-y_{ip}\right|$$
where n is the size of the training data set, k is the number of parameters for the model, y_i_ is the actual value of the response variable (BW), y_ip_ is the predicted value for the response variable (BW), S_d_ is the standard deviation for the response variable (BW), and S_m_ is the standard deviation for the optimum model’s errors.


RMSE, SDR, CV, PI, RAE, MAPE, MAD, R^2^ and AIC goodness-of-fit criteria were used to evaluate the performance of the model. The model performances were evaluated according to the lowest RMSE, SDR, PI, RAE, MAPE, MAD and AIC values and the highest R^2^ value [21].

Statistical evaluations were carried out using the R software. To provide information about the structure of the data, descriptive statistics were performed. Descriptive statistics for all variables were estimated using “psych” package in the R environment [22]. The “caret” packages in the R software were used to analyze the MARS algorithm [23]. To evaluate the MARS model performances, the “ehaGoF” package was employed [24].

## 3. Results

The descriptive statistics of response and explanatory variables for Anatolian buffaloes according to the sex (female and male) factor, reported as mean ± standard error, are given in Table 1. To compare the sex factor for each variable, the two-sample t test was performed.

Table 2 shows the Pearson’s correlation coefficients for determining the relationship between BW and body measurements. Body weight had a significant correlation with the TL (0.850), SH (0.538), WH (0.527), BL (0.640), CC (0.845), SD (0.835) and BW (0.153), respectively. A high correlation coefficient was observed between CC and BW 0.845 (*p* < 0.01). Another strong correlation was recorded at the *p* < 0.01 level for CC-TL (0.837) and SD-BW (0.835).

The model performance results for different training and test sets, such as 65:35, 70:30 and 80:20, based on the goodness-of-fit criteria, are given in Table 3. According to Table 3, the best predictive model was achieved for the proportion 70–30%. The proportion 70–30% had the smallest RMSE, SDR, CV, PI, RAE, MAPE, MAD and AIC values. Additionally, the highest R^2^ value was determined for the proportion 70–30%. In addition, the Pearson’s correlation coefficient was determined to be 0.93 and 0.905 for the actual data and predicted data for training and test sets, respectively.

For the training and testing set proportion of 70–30%, the results for the obtained MARS model with five terms and degree: 1 are given in Table 4.

According to Table 4, the first term of the model was an intercept that had a coefficient of 122.710. The second term, TL, had a cutpoint of 98 cm for a negative coefficient of 1.767. The third term, BL, had a cutpoint of 96 cm with a coefficient of 1.098. The fourth term was CC, with a coefficient of 1.824. The last term was SD, with a coefficient of −3.766.

Within the scope of estimating body weight, the optimal MARS estimation model enables breeders to make more accurate decisions regarding herd management, such as the appropriate feed amount, drug doses in buffaloes and determining the marketing price per animal. This may also enable them to reveal morphological features that positively affect body weight for use as indirect selection criteria.

## 4. Discussion

Body weight estimation methods based on body measurements are widely used to determine the relationship between the structures of animal species. In addition to using body measurements to estimate BW, the validity of the statistical method used is also important. In the literature, there are no studies in which live weight is estimated from the body measurements of buffaloes, even though the importance of weight estimation is well-known [25,26]

Male Anatolian buffaloes had higher means for CC, SD, BW and KG in comparison to female buffaloes (Table 1) (*p* < 0.01). Husni et al. [27] determined the optimal regression model with BW, BL, HW and HG for the Doro Ncanga buffalo. They reported that the correlation between BW and BL, HW and HG was 0.319, 0.071 and 0.967, respectively. The present study found that the correlation coefficient between BL, WH and CC was 0.640, 0.527 and 0.845, respectively. Kelgökmen and Ünal [28] reported that the correlation was 0.64 between Bl and SH and 0.71 between L and CC, whereas our study found this was 0.38 and 0.56. Our results show lower coefficients than those found by [28]; this may be the result of sample size because Kelgökmen and Ünal [28] used 73 animals. In addition, within the scope of regression analysis, the models had an R^2^ value between 0.001 and 0.957. Our results show R^2^ values of 0.864 and 0.810 for the training and test sets, respectively. These differences may be due to the different buffalo breeds studied.

Johari et al. [29] determined the best regression equation by using the body weight, body length, shoulder height, pelvic height, chest depth, chest circumference, chest width, pelvic width and waist width within the scope of multiple linear regression procedures. For this aim, they used swamp buffaloes. They reported that the correlation coefficients between BW and body length, shoulder height and chest circumference were 0.896, 0.776 and 0.935, respectively. In the present study, the correlation coefficient was lower than that found by Johari et al. [29]. However, the regression equation included the same parameters, body length and chest circumference, and excluded chest depth, to determine body weight. In addition to these variables, TL and SD were determined to be important parameters according to the MARS algorithm in the present study.

Due to the scarcity of such studies on buffaloes, we discuss different animal species. Celik [30] estimated live body weight with some body measurements in Pakistani goats and emphasized the superiority of the MARS estimation model with 0.91 Rsq, 0.86 ARsq, 3.32 RMSE, 0.30 SDR and 8.49 MAPE. Celik and Yilmaz [31] recorded 0.845 Rsq, 0.828 ARsq, 0.393 SDR, 2.893 RMSE and 5.047 MAPE for the MARS estimation model in estimating the body weight of Kars Shepherd dogs. Tırınk [32] compared Bayesian Regularized Neural Networks, Random Forest Regression, Support Vector Regression and Multivariate Adaptive Regression Splines Algorithms to predict body weight from biometrical measurements in Thalli sheep and mentioned that the MARS algorithm was more recommendable according to root mean square error (RMSE), standard deviation ratio (SDR), performance index (PI), global relative approximation error (RAE), mean absolute percentage error (MAPE), Pearson’s correlation coefficient (r), determination of coefficient (R^2^) and Akaike’s information criteria (AIC). The difference can be attributed to the animal species. Çanga [33] mentioned that the MARS algorithm could allow livestock breeders to obtain effective clues by using independent variables such as breed, age and body weight in estimating hot carcass weight with the determination coefficient of 0.975. Şengül et al. [34] mentioned that MARS and Bagging MARS algorithms revealed correct results according to the goodness-of-fit statistics, and it has been determined that the MARS algorithm gives better results in live weight modeling. Tariq et al. [35] used multiple linear regression to estimate BW from body measurements and body condition scoring for Nili-Ravi buffaloes; as a result, they mentioned that the multiple linear regression between BW and heart girth, body length and body condition scores was significantly higher with a determination coefficient of 0.95. However, they did not mention anything about the curve estimation, which should be included. Iqbal et al. [36] compared nonlinear functions (Gompertz, logistic, negative exponential, Brody and Bertalanffy) and the MARS algorithm for modeling and predicting the growth of indigenous Harnai sheep. As a comparison criterion, they used the adjusted coefficient of determination (R^2^adj), Durbin–Watson statistic (DW), root mean square error (RMSE), Akaike’s and Bayesian information criteria (AIC and BIC) and the coefficient of correlation (r) between observed and fitted live body weight. As a result, [36] argued that the MARS algorithm could be used more reliably for prediction. Faraz et al. [37] compared CART and MARS algorithms to predict live body weight based on body measurements in Thalli sheep; they mentioned that the MARS algorithm was superior to CART according to the comparison criteria used. Tırınk et al. [10] used the MARS algorithm to predict body weight from body measurements in Marecha (Camelus dromedaries) camels. They mentioned that the best MARS model for BW prediction was obtained using sex and shoulder height as independent variables for an 80:20 training and test set proportion. The MARS algorithm had better identification properties than other prediction models when compared theoretically.

## 5. Conclusions

In our study, in which we performed the estimation of live weight with the MARS model using different ratios of training and test sets, it was concluded that the MARS algorithm is very successful in estimating the live weight of buffaloes from body measurements. In addition, the results obtained with the MARS algorithm provide us with the appropriate herd management conditions. This situation is thought to help with indirect selection criteria. The semiautomatic devices that are available are expensive for farmers in remote areas, whereas simple body measures can be performed by every farmer.

## Figures and Tables

**Table 1 animals-12-02923-t001:** Descriptive statistics.

Variables	Female (Mean ± Standard Error)	Male (Mean ± Standard Error)	Probability (P)
TL	101.52 ± 0.62	103.19 ± 0.59	0.07
SH	109.34 ± 0.70	109.57 ± 1.53	0.88
WH	106.67 ± 1.20	109.50 ± 0.67	0.07
BL	103.91 ± 0.90	105.42 ± 1.62	0.38
CC	126.06 ± 1.11 b	129.64 ± 1.09 a	0.02
SD	23.06 ± 0.20 b	24.09 ± 0.21 a	0.00
BiW	28.95 ± 0.22 b	30.31 ± 0.34 a	0.00
BW	129.21 ± 2.87 b	141.44 ± 3.29 a	0.00

TL: tail length, SH: shoulder height, WH: withers height, BL: body length, CC: chest circumference, SD: shank diameter, BiW: birth weight, BW: body weight. a, b: different letters in same row (between genders) shows statistical difference (*p* < 0.05).

**Table 2 animals-12-02923-t002:** Correlation matrix.

	TL	SH	WH	BL	CC	SD	BiW	BW
TL	1.000							
SH	0.580	1.000						
WH	0.582	0.401	1.000					
BL	0.603	0.377	0.355	1.000				
CC	0.837	0.568	0.532	0.561	1.000			
SD	0.739	0.477	0.445	0.629	0.788	1.000		
BiW	0.119	0.105	0.108	0.018	0.148	0.210	1.000	
BW	0.850	0.538	0.527	0.640	0.845	0.835	0.153	1.000

**Table 3 animals-12-02923-t003:** Goodness-of-fit criteria results for each model.

Criterion	65–35%		70–30%		80–20%	
	Train	Test	Train	Test	Train	Test
Root mean square error (RMSE)	11.499	13.447	10.996	12.675	9.416	14.896
Standard deviation ratio (SDR)	0.393	0.483	0.369	0.46	0.333	0.481
Coefficient of variation (CV)	8.58	10.09	8.15	9.06	7.04	11.13
Performance index (PI)	4.452	5.352	4.205	5.061	3.609	5.891
Relative approximation error (RAE)	0.007	0.01	0.006	0.009	0.005	0.012
Mean absolute percentage error (MAPE)	6.352	7.244	6.077	7.363	5.367	8.121
Mean absolute deviation (MAD)	8.654	9.916	8.335	9.748	7.105	11.345
Coefficient of determination (R^2^)	0.845	0.765	0.864	0.810	0.889	0.765
Akaike’s information criterion (AIC)	557.072	316.654	594.997	258.882	637.392	186.87

**Table 4 animals-12-02923-t004:** MARS model.

Variables	Coefficients
Intercept	122.710
h (98-TL)	−1.767
h (BL-96)	1.098
h (CC-122)	1.824
h (26-SD)	−3.766

## Data Availability

For data requests, please contact the author H.Ö.

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
