# Peer review of "Use of Multivariate Adaptive Regression Splines Algorithm to Predict Body Weight from Body Measurements of Anatolian buffaloes in Türkiye"

_animals, 2022, doi:10.3390/ani12212923_

Round 1
Reviewer 1 Report
The source part should be expanded a little more. More recent sources should be added.
Author Response
First of all, the authors thank reviewers for their valuable comments that improve the quality of this manuscript.
Reviewer Comment
The source part should be expanded a little more. More recent sources should be added.
Answer
According to the reviewer's wish, seven related new articles were added throughout the text.

Reviewer 2 Report
Reviewed manuscript "Use of multivariate adaptive regression splines algorithm to predict of body weight from body measurements of Anatolian buffaloes in Türkiye" (animals-1904503) contains the results of interesting research work of scientific and practical significance. Anatolian Buffalo is an important breed reared for meat and milk yield in various regions of Türkiye. The present study was performed to estimate body weight from some body measurements. The most considered are physiological growth traits such as live body weight and gain can be a good and easy breed tool.
The experiment was planned properly and carried out on sufficiently numerical material animals kept on the professional farm.
Statistical analysis of the obtained results is correct.
Tables presented the results and statistical data were constructed properly.
The discussion was carried out properly and the literature used in this part of the manuscript was chosen accordingly.
Authors cited only 25 references items, it would be good to supplement them with additional ones.
In summary - the manuscript contains valuable results and should be publishing in Animals after the corrections.
Author Response
First of all, the authors thank the reviewers for their valuable comments that improve the quality of this manuscript.
Reviewer Comment
First of all the authors thanks to reviewer’ praise of manuscript.
Authors cited only 25 references items, it would be good to supplement them with additional ones.
Answer
According to the reviewer's wish, seven related new articles were added throughout the text.

Reviewer 3 Report
I am afraid I am not understanding the value of this paper. Maybe I am wrong, but there are now semi-automated techniques to estimate weight of farm animals that are more advanced than what the authors are presenting. The authors present only evidence that several body measurements are linked to body weight, and there are papers from the 1950s that show this link. They did a predictive model and not just a correlation, but still I would have expected to compare different predictive models to make the paper stronger. I disagree that there are few papers on this topic, and I do not see what this paper can add to the current literature. The paper is also lacking details and references. The narrative of the paper is quite clear but significance and originality are low in my opinion.
Author Response
First of all, the authors thank reviewers for their valuable comments that improve the quality of this manuscript.
Reviewer Comment
I am afraid I am not understanding the value of this paper. Maybe I am wrong, but there are now semi-automated techniques to estimate weight of farm animals that are more advanced than what the authors are presenting. The authors present only evidence that several body measurements are linked to body weight, and there are papers from the 1950s that show this link. They did a predictive model and not just a correlation, but still I would have expected to compare different predictive models to make the paper stronger. I disagree that there are few papers on this topic, and I do not see what this paper can add to the current literature. The paper is also lacking details and references. The narrative of the paper is quite clear but significance and originality are low in my opinion.
Answer
The reviewer is right that “there are now semi-automated techniques to estimate weight of farm animals that are more advanced than what the authors are presenting”, but there is no such study made for Anatolian buffaloes with MARS algorithm.
There are many thousands of articles produced each year with using old aims with new techniques.
Asking to compare different predictive models to make the paper stronger is good advice, but as a skilful biometrician I can easily express that in such complicated methods it is clearer to explain model lonely. In the future studies we will take into account the reviewer’ this comment.
Indeed there are many papers estimating weight by using biometrical properties but this is the new study using MARS algorithm on Anatolian buffaloes, this is the originality because each species or breed need individual methods.
According to the reviewer's wish, seven related new articles were added throughout the text.
